# Fabrication of a Novel AgBr/Ag_2_MoO_4_@InVO_4_ Composite with Excellent Visible Light Photocatalytic Property for Antibacterial Use

**DOI:** 10.3390/nano10081541

**Published:** 2020-08-06

**Authors:** Jie Zhang, Jia Wang, Qingjun Zhu, Binbin Zhang, Huihui Xu, Jizhou Duan, Baorong Hou

**Affiliations:** 1CAS Key Laboratory of Marine Environmental Corrosion and Bio-fouling, Institute of Oceanology, Chinese Academy of Sciences, Qingdao 266071, China; wangjia2@qdio.ac.cn (J.W.); zhuqingjun@qdio.ac.cn (Q.Z.); xuhuihui16@mails.ucas.ac.cn (H.X.); duanjz@qdio.ac.cn (J.D.); brhou@qdio.ac.cn (B.H.); 2Open Studio for Marine Corrosion and Protection, Pilot National Laboratory for Marine Science and Technology, Qingdao 266237, China; 3University of Chinese Academy of Sciences, Beijing 100049, China; 4Center for Ocean Mega-Science, Chinese Academy of Sciences, 7 Nanhai Road, Qingdao 266071, China

**Keywords:** photocatalyst, organic pollutants, antibacterial, Z-type photocatalytic mechanism

## Abstract

A novel AgBr/Ag_2_MoO_4_@InVO_4_ composite photocatalyst with different heterojunction structures was successfully constructed by compounding InVO_4_ with Ag_2_MoO_4_ and AgBr. According to the degradation, antibacterial and free radical trapping data, the photocatalytic antibacterial and antifouling activities of AgBr/Ag_2_MoO_4_@InVO_4_ composite were evaluated, and the corresponding photocatalytic reaction mechanism was proposed. Adding AgBr/Ag_2_MoO_4_@InVO_4_ composite, the degradation rate of ciprofloxacin (CIP) achieved 95.5% within 120 min. At the same time, the antibacterial rates of *Escherichia coli* (*E. coli*), *Staphylococcus aureus* (*S. aureus*) and *Pseudomonas aeruginosa* (*P. aeruginosa*) achieved 99.99%. The AgBr/Ag_2_MoO_4_@InVO_4_ composite photocatalyst showed promising usage in photocatalytic antibacterial and purification areas.

## 1. Introduction

In recent years, new green photocatalysis technology based on semiconductors has developed rapidly. It has been widely used in pollutant degradation, water separation and sterilization through using solar energy as an energy source [1,2]. Semiconductor photocatalysis technology has attracted more and more attention due to its energy efficiency, simple operation, low expense and high stability, and because it is also green and non-toxic, it has no secondary pollution, among other advantages [3,4,5]. In order to use sunlight effectively, it is necessary to design new photocatalytic heterojunction materials with a visible light band response. To date, a number of excellent semiconductor photocatalysts have been developed rapidly, such as metal oxides [6,7,8], metal sulfides [7,9,10], metal oxynitrides [11] and polymer materials [12]. InVO_4_ is a metal vanadate photocatalyst with a band gap width of 2.0 eV, and it has excellent photocatalytic performance under visible light [13,14]. InVO_4_ has gained widespread attention in many fields, such as degradation, air purification, water decomposition, organic pollutants, etc. However, the competence of the InVO_4_ photocatalyst is largely affected by its size and micro-morphology, resulting in low efficiency of photogenerated carriers [15]. Use of a supporting cocatalyst on the surface of the photocatalyst is considered as one of the most effective methods to drive the separation of photogenerated electrons and holes and enhance photocatalytic activity [15,16]. Ag_2_MoO_4_ has many advantages such as excellent antibacterial activity, high conductivity, good electrochemical energy storage, suitable valence band position, generation of a large number of hydroxyl radicals, controllable morphology, photoluminescence, etc. [17,18]. AgBr has been extensively studied as a promoter of Ag-based semiconductors. Similar studies, such as with Ag/AgBr/AgVO_3_ [19], Ag/AgBr–Bi_2_MoO_6_ [20], Ag@AgBr/CaTiO_3_ [21] and Ag/AgBr@InVO_4_ [22], have also confirmed that AgBr is promising to prepare efficient and stable photocatalysts. 

As InVO_4_, Ag_2_MoO_4_ and AgBr have their own advantages and disadvantages. Many researchers combined them with other substances to learn their strengths and make up for their shortcomings, and then they prepared photocatalysts with better performance for various studies. Chen et al. [23] fabricated a new β-Ag_2_MoO_4_/BiVO_4_ heterojunction photocatalyst by a simple precipitation method at room temperature. The degradation rates for rhodamine B (RhB) of pure β-Ag_2_MoO_4_ and BiVO_4_ were not good, but the as-created β-Ag_2_MoO_4_/BiVO_4_ photocatalyst had about 92.6% RhB degradation rate. The combination of Ag_2_MoO_4_ and BiVO_4_ increased the absorption of visible light, improved the transfer speed of photogenerated electrons and reduced the recombination of holes and electrons, and it had excellent photocatalytic performance. Yang et al. [24] constructed an InVO_4_/β-AgVO_3_ nanocomposite photocatalyst by a facile hydrothermal method and subsequent in situ growth process. The as-prepared InVO_4_/β-AgVO_3_ composite photocatalyst had an enhanced photocatalytic performance in reducing CO_2_ to CO under visible light. Li et al. [25] reported a g-C_3_N_4_/graphene oxide-Ag/AgBr composite photocatalyst used to prepare hydrogen. Due to the synergistic effect of silver bromide, with good photosensitivity, and silver plasma, the photocatalyst improved the hydrogen evolution performance and provided a feasible method for developing hydrogen energy.

To date, there are no reports on the composite photocatalyst of AgBr/Ag_2_MoO_4_@InVO_4_. In this work, a hydrothermal method and in situ growth method were used to produce AgBr/Ag_2_MoO_4_@InVO_4_ photocatalytic composites with different molar ratios. X-ray diffraction (XRD), scanning electron microscopy (SEM), energy-dispersive spectroscopy (EDS), energy-dispersive X-ray spectroscopy (EDX) and high-resolution transmission electron microscopy (HRTEM) were used to characterize the microstructure and composition of the prepared composite photocatalyst. Using visible light as a light source, the photocatalytic degradation of the organic pollutant ciprofloxacin (CIP) was tested. The degradation rates of different molar ratios of AgBr/Ag_2_MoO_4_@InVO_4_ photocatalyst to CIP solution were calculated under the same test conditions. At the same time, *E. coli*, *S. aureus* and *P. aeruginosa* were selected as model bacteria to carry out antibacterial experiments on the prepared AgBr/Ag_2_MoO_4_@InVO_4_ photocatalytic composite materials in order to study the bactericidal performance of the photocatalyst. In addition, the photocatalytic reaction mechanism of AgBr/Ag_2_MoO_4_@InVO_4_ heterojunction was proposed based on free radical trapping experiments, degradation and sterilization data.

## 2. Experimental Section

### 2.1. Synthesis of AgBr/Ag_2_MoO_4_@InVO_4_ Photocatalysts

All chemical reagents in our experiments were analytical reagent grade. In a classic order for the preparation of InVO_4_ materials, 0.117 g NH_4_VO_3_(Shanghai, China) was dissolved in 50 mL water firstly. After that, it was sonicated and stirred continuously for 20 min at normal room temperature to get a homogeneous solution. After dissolving and stirring 0.382 g of In(NO_3_)_3_(Shanghai, China) in 10 mL water, this liquid was added dropwise slowly to the former solution. The pH value was controlled to 4.0 using 0.25 wt. % NH_3_·H_2_O (Shanghai, China) and 2 mol/L HNO_3_(Shanghai, China). The mix solution was stirred in succession for 30 min until a yellow colloidal solution was acquired. The mix solution was moved to a 100mL Teflon-lined stainless-steel autoclave, which had been heated to 200 °C for 24 h. Having been cleaned many times using ultrapure water and absolute ethyl alcohol, the obtained solid composite materials were centrifuged to obtain yellow solid powders. Finally, the composite materials were dried at 60 °C for 6 h.

The 0.2068 g InVO_4_ semiconductor material prepared above was put into 30 mL distilled water and treated ultrasonically for 30 min. After that, 0.238 g AgNO_3_ (Shanghai, China) was added into the above solution and mixed for 30 min to make them disperse evenly. Subsequently, 0.164 g of cetyltrimethylammonium bromide (CTAB, Beijing, China) and 0.109 g of Na_2_MoO_4_.2H_2_O (Shanghai, China) were dissolved in 20 mL distilled water. Next, this solution was slowly added into the above solution drop by drop. The reaction mixture was stirred continuously for 2 h in a dark environment. The composite material gained was cleaned three times using water and ethanol respectively. Then, the precipitates were dried for 12 h in an oven at 60 °C. The prepared sample was marked as “1.0AgBr/Ag_2_MoO_4_@InVO_4_”. In this way, composite materials with different molar ratios were prepared and labeled as X AgBr/Ag _2_MoO_4_ @ InVO_4_ (x = 0.2, 0.6, 1.0 and 1.4).

### 2.2. Characterization

XRD (Rigaku D/max-3C, Tokyo, Japan) was used to characterize the crystalline structure of the samples. The microstructure of the prepared photocatalyst was examined by SEM (Hitachi S-4800, Tokyo, Japan), TEM (Tecnai G2F20, Hillsboro, OR, USA) and HRTEM (FEI Company, Oregon, USA). A UV–vis spectrophotometer (U-2900, Tokyo, Japan) was used to characterize the absorption spectra, and ultra-pure water was used as the reference.

### 2.3. Photocatalytic Performance

In this experiment, ciprofloxacin (CIP), an organic substance, was chosen as a model molecule to judge the photocatalytic degradation property of the material. Before the reaction, condensed water was acquired, and an 800W Xe lamp (XPA-7, Xujiang Electromechanical Plant, Nanjing, China) deployed with a 420 nm cut-off filter was turned on. Then, 40 mg photocatalyst was put into 50 mL CIP solutions, and quartz tubes were inserted into the photochemical reactor filled with condensed water. In the photocatalytic reaction process, the solution was blended magnetically in the dark for 30 min so CIP and the synthetic materials were well-distributed. After the light shield had been pulled up, the sample solution extracted in the same time interval was filtered through a membrane to eliminate solid particles. The residual amount of CIP in the extracted solution was measured by an ultraviolet–visible spectrometer (Hitachi U-2900, Hitachi, Tokyo, Japan). The amount of CIP was determined by comparing the peak-to-peak value of the sample between the standard sample. In the measurement experiment, ultra-pure water was used as a reference, and the scanning range was 200–700 nm. 

In this experiment, *S. aureus, E. coli* and *P. aeruginosa* were selected to evaluate the antibacterial performance of the photocatalyst. An 800W Xe lamp using a 420 nm cut-off filter was adopted as the light source. Typically, 45 mL phosphate-buffered saline (PBS), 30 mg photocatalyst and 5 mL bacterial suspension were added into 50mL quartz tubes. The mingled liquids were stirred for 30 min using a magnetic stirrer in total darkness to balance the adsorption/desorption. During 800W Xe lamp irradiation, 2 mL of mixed solution was taken out every 20 min and diluted with PBS in different gradients. Next, LB agar plates were used for cultivating the diluted bacterial suspension at the temperature of 37 °C for 24 h. Then, the number of bacteria was calculated via plate counts. In each group, the survival rate and antibacterial rate were calculated through triplicate parallel experiments.

The survival rate was calculated by the formula [26]:Survival rate (%) = N_t_/N_0_ × 100,(1)

Among them, N_0_ and N_t_ are the number of bacteria in the blank group and the number of bacteria in the antibacterial experiment, respectively. The formula of antibacterial rate [5] is
Antimicrobial rate (%) = 100-survival rate(2)

At present, many researchers have confirmed that free radical active substances (∙OH, ∙O_2_^−^, h^+^, etc.) play a major role in photocatalytic reactions. In this experiment, free radical trapping experiments were used to study the types of free radicals. In this experiment, isopropyl alcohol (IPA), p-benzoquinone (BQ) and sodium oxalate (MSDS) were used as the ∙OH capture agent, ·O_2_^−^ capture agent and h^+^ capture agent, respectively. The photocatalytic reaction mechanism was studied in combination with the above experiments. The operation steps were consistent with the photocatalytic degradation experiment. Then, the degradation rate was calculated. The active species were also analyzed.

## 3. Results and Discussion

### 3.1. Characterization of AgBr/Ag_2_MoO_4_@InVO_4_

The XRD patterns of InVO_4_ crystals and AgBr/Ag_2_MoO_4_@InVO_4_ photocatalytic composites are shown in Figure 1. The characteristic peaks appeared at 18.6°, 20.8°, 23.0°, 24.9°, 27.1°, 31.1°, 33.1°, 35.2° and 47.0°, which verified the presence of the monoclinic InVO_4_ phase (JCPDS No.48−0898) for the (1 1 0), (0 2 0), (1 1 1), (0 2 1), (2 0 0), (1 1 2), (1 3 0) and (2 2 2) planes, respectively [13,27]. The peaks at 2θ equal to 27.4°, 32.8° and 37.6° of AgBr/Ag_2_MoO_4_@InVO_4_ composites corresponded to Ag_2_MoO_4_ (JCPDS No.21–1340) for the (2 1 2), (3 1 0) and (3 2 0) planes, respectively. The diffraction peaks at 2θ equal to 44.33°, 55.04° and 73.24° were assigned to (2 2 0), (2 2 2) and (4 2 0) planes of AgBr [28,29]. The diffraction peaks of AgBr corresponded to the JCPDS card No.79-0149. The XRD pattern of AgBr/Ag_2_MoO_4_@InVO_4_ composite materials showed strong InVO_4_ and Ag_2_MoO_4_ diffraction peaks. Comparing the InVO_4_ pattern with AgBr/Ag_2_MoO_4_@InVO_4_ pattern, the intensity of the InVO_4_ peak declined mildly with the addition of AgBr and Ag_2_MoO_4_, proving that AgBr/Ag_2_MoO_4_ particles were fixed on the surface of InVO_4_.

In AgBr/Ag_2_MoO_4_@InVO_4_ samples, no obvious change in the diffraction peak position of InVO_4_ was observed, which means that the introduction of AgBr/Ag_2_MoO_4_ did not destroy the crystal structure of InVO_4_. In addition, AgBr/Ag_2_MoO_4_@InVO_4_ composite material had no other miscellaneous peaks. Therefore, it could be proved that the AgBr/Ag_2_MoO_4_@InVO_4_ composite was correctly prepared. 

The morphology of the composite photocatalyst was observed by SEM. SEM images of 1.0Ag_2_MoO_4_@InVO_4_ are shown in Figure 2a–c, showing a clear waxberry-like structure with an average diameter of 8 μm. Some polygonal grains could also be clearly found on the surface. The resultant SEM images (Figure 2d–f) of 1.0AgBr/Ag_2_MoO_4_@InVO_4_ composite indicated that the spherical morphology of InVO4 was saved. Furthermore, AgBr and Ag_2_MoO_4_ particles were uniformly distributed on the InVO_4_ surface forming AgBr/Ag_2_MoO_4_@InVO_4_ heterostructures. To some extent, the AgBr/Ag_2_MoO_4_@InVO_4_ heterostructures could enhance the specific surface area of the material and offer more active sites for photocatalytic reaction.

The morphologies of 1.0AgBr/Ag_2_MoO_4_@InVO_4_ composite were further illustrated by TEM and HRTEM. The amplified TEM images in Figure 3a–b distinctly display the anomalistic particles around the microsphere circumambience, proving that the AgBr/Ag_2_MoO_4_@InVO_4_ particles were tightly bound together. Figure 3c exhibits the HRTEM image of 1.0AgBr/Ag_2_MoO_4_@InVO_4_, in which three sets of different crystal streaks were observed expressly. Consequently, these results further demonstrate that a well-defined heterojunction structure had taken shape between AgBr, Ag_2_MoO_4_ and InVO_4_. Moreover, EDS measurements revealed the elemental composition of AgBr/Ag_2_MoO_4_@InVO_4_ ternary composites, which provided direct evidence for the coexistence of AgBr, Ag_2_MoO_4_ and InVO_4_. As shown in Figure 3d, the above AgBr/Ag_2_MoO_4_@InVO_4_ composite was composed of O, Br, Mo, Ag, In and V, illustrating the formation of AgBr/Ag_2_MoO_4_@InVO_4_ composite photocatalyst including Ag_2_MoO_4_, AgBr and InVO_4_. In addition, SEM elemental mapping described the composition and distribution of the different elements. As shown in the elemental mapping images (EMIs), the five Ag, Br, In, Mo and V elements (Figure 3e) could be observed existing homogeneously within the selected area on the AgBr/Ag_2_MoO_4_@InVO_4_ photocatalyst.

### 3.2. Photocatalytic Property Study

Ciprofloxacin (CIP), as an antibiotic, protects people’s health, but it also causes some environmental pollution. Using a photocatalyst to degrade CIP has low cost and no secondary pollution. Photocatalysts can generate photo-generated carriers under certain light illumination, and the photo-generated carriers can react with water to generate active hydroxyl groups (∙OH) and superoxide radicals (∙O_2_^−^), which can decompose CIP into small molecular inorganic substances. In addition, the generated holes (h^+^) can also oxidize CIP directly or indirectly [30]. The photocatalytic decontamination performances of the obtained AgBr/Ag_2_MoO_4_@InVO_4_ samples were appraised according to the CIP degradation rate in visible light [5]. As demonstrated in Figure 4a, under the condition of no photocatalyst, no degradation of CIP was observed, indicating that CIP was stabilized in visible light. In visible light irradiation, the degradation rate of InVO_4_ for CIP was lower than that of composite AgBr/Ag_2_MoO_4_@InVO_4_. For composite materials, the degradation efficiency of AgBr/Ag_2_MoO_4_@InVO_4_ for CIP was above 95.5%, and the degradation efficiency of 1.0 AgBr/Ag_2_MoO_4_@InVO_4_ for CIP was the highest. In Figure 4, a linear relationship between -ln(C/C_0_) and reaction time (T) is displayed, which indicates that the reaction process conformed to the pseudo-first-order reaction kinetic process [31].
−ln (C/C_0_) = K_app_ T(3)

C is CIP concentration with reaction time T, C_0_ is initial CIP concentration, and K_app_ is the apparent rate constant. According to the above formula, the K_app_ of InVO_4_, 0.2 AgBr/Ag_2_MoO_4_@InVO_4_, 0.4 AgBr/Ag_2_MoO_4_@InVO_4_, 1.0 AgBr/Ag_2_MoO_4_@InVO_4_ and 1.4 AgBr/Ag_2_MoO_4_@InVO_4_ are 0.0006, 0.024, 0.02, 0.037 and 0.029 min^−1^ respectively. Notably, compared with other photocatalysts, the 1.0 AgBr/Ag_2_MoO_4_@InVO_4_ composite photocatalyst had the strongest CIP degradation activity. In addition, compared with other photocatalysts such as InVO_4_/ZnFe_2_O_4_ [32] and Ag/AgCl/BiOCOOH [30], the 1.0 AgBr/Ag_2_MoO_4_@InVO_4_ composite photocatalyst showed better ability to degrade organic pollutants.

A photocatalyst can not only degrade organic pollutants, but it can also kill bacteria in sewage. Many researchers believe that photocatalysts can produce holes (h^+^) and other active substances under the action of light, which can interact with bacterial cell membranes and affect the permeability of cell membranes, leading to disorder of bacterial physiological processes and eventually leading to bacterial death [33]. In this paper, *E. coli* (9.8 × 10^6^ cfu/mL), *S. aureus* (2.1 × 10^6^ cfu/mL) and *P. aeruginosa* (2.7 × 10^6^ cfu/mL) were selected to study the photocatalytic antibacterial activity of photocatalysts in visible light [13]. As shown in Figure 5a, it can be seen from the survival curve of *P. aeruginosa* that the number of *P. aeruginosa* did not noticeably change in blank control experiments. This revealed that the influence of visible light and the toxicity of the photocatalyst itself on bacterial activity could be ignored. In addition, as can be seen from Figure 5a, for *P. aeruginosa*, the 1.0AgBr/Ag_2_MoO_4_@InVO_4_ composite had better antibacterial activity than pure InVO_4_ and other molar ratios of AgBr/Ag_2_MoO_4_@InVO_4_ composites. In addition, as shown in Figure 5b, under the photocatalytic condition of 1.0AgBr/Ag_2_MoO_4_@InVO_4_ composite after 60 min, the sterilization rates of *E. coli, S. aureus* and *P. aeruginosa* were 99.9999%, 99.9998% and 99.9997%, respectively, indicating that the catalyst had higher antibacterial and antifouling activity. In addition, compared with other reported antifouling photocatalysts such as g-C_3_N_4_@Ag/AgVO_3_ [34], AgBr/TiO_2_/graphene aerogel [35], AgI/BiVO_4_ [36] and InVO_4_/AgVO_3_ [27], the 1.0 AgBr/Ag_2_MoO_4_@InVO_4_ composite photocatalyst in this experiment showed quite outstanding photocatalytic antibacterial performance, revealing potential application value in sterilization and marine antifouling.

Based on the research of many scholars, it could be concluded that free radical active species play a vital role in photocatalytic reactions. To further prove the influence of free radical active substance on the photocatalytic reaction, we conducted a free radical capturing experiment. Isopropyl alcohol (IPA), p-benzoquinone (BQ) and sodium oxalate (MSDS) served as the ∙OH capture agent, ∙O_2_^−^ capture agent and h^+^ capture agent, respectively [16,37]. As shown in Figure 6, 94.9% of ciprofloxacin (CIP) was degraded without the capture agent after 120 min illumination. After adding 1 mmol IPA, the degradation rate of CIP decreased to 93.7%. Moreover, after adding 1 mmol BQ and 1 mmol MSDS, the antibacterial rates decreased to 30.0% and 31.3%, indicating that the photocatalytic performance of 1.0AgBr/Ag_2_MoO_4_@InVO_4_ was significantly inhibited. Therefore, these experiment results proved that ∙O_2_^−^ and h^+^ played a crucial role in photocatalytic degradation of CIP. To sum up, we can conclude that the main active substances of the AgBr/Ag_2_MoO_4_@InVO_4_ photocatalyst for CIP degradation were ∙O_2_^−^ and h^+^. CIP was oxidized by ∙O_2_^−^ and h^+^ generated by the photocatalyst into a small molecular product. The possible reaction process can be shown as follows [38]:
photocatalyst + *hv* → e^−^ + h^+^(4)
O_2_ + e^−^ →∙O_2_^−^(5)
CIP + h^+^ + ∙O_2_^−^ → CO_2_ + small molecules(6)

A Z-type photocatalytic mechanism of composite AgBr/Ag_2_MoO_4_@InVO_4_ was proposed, based on the analysis of experimental results. The energy band structures of AgBr, InVO_4_ and Ag_2_MoO_4_ as well as the degradation of CIP and sterilization mechanism of AgBr/Ag_2_MoO_4_@InVO_4_ photocatalyst are shown in Figure 7. When the photon energy was greater than the bandgap, electrons (e^−^) in the valence band of AgBr (with narrow band gap) were excited to the guide band easily, and photoelectrons and holes (h^+^) appeared in visible light [39]. At the same time, electrons (e^−^) on the AgBr conduction band (−0.30 eV) could transfer to Ag_2_MoO_4_ (−0.18 eV) readily. At this time, the electrons (e^−^) on the Ag_2_MoO_4_ conduction band quickly moved to the valence band of AgBr through the heterojunction and newly combined with the holes (h^+^) on the valence band of AgBr [4,5]. In addition, holes (h^+^) in the valence band of InVO_4_ [13] could also be transferred to the conduction band of AgBr, which characterizes the band potential difference. A Z-type mechanism was set up. Electrons (e^−^) in the conduction band (−0.57 eV) of InVO_4_ had very strong reduction performance [27], while holes (h^+^) in the valence band (3.02 eV) of Ag_2_MoO_4_ had good oxidation capability. The Z-type structure effectively separated electrons (e^−^) and holes (h^+^) and enhanced the photocatalytic capability of AgBr/Ag_2_MoO_4_@InVO_4_ composite. InVO_4_ has a more negative charge conducting potential (−0.57 eV) than E^0^ (O_2_/∙O_2_^−^) (−0.046 eV vs. Normal Hydrogen Electrode (NHE)), which can produce the active substance ∙O_2_^−^ [40]. These free radicals can not only oxidize and degrade CIP, but they also have bactericidal effects. The mechanism showed that photocatalytic antibacterial antifouling technology has the advantages of high efficiency, environmental protection and no secondary pollution.

## 4. Conclusions

In this paper, a new AgBr/Ag_2_MoO_4_@InVO_4_ photocatalytic composite with a microsphere-like morphology was produced successfully by hydrothermal and in situ growth methods, and the photocatalytic antibacterial activity was determined. The chemical composition and morphology of the AgBr/Ag_2_MoO_4_@InVO_4_ photocatalytic composites were proved by XRD, SEM, EDS, EDX and HRTEM. Under visible light, the photocatalytic experiments showed that the 1.0 AgBr/Ag_2_MoO_4_@InVO_4_ photocatalytic composite had a much higher photocatalytic performance compared to pure InVO_4_ and other AgBr/Ag_2_MoO_4_@InVO_4_ composites. Furthermore, the antibacterial activity of this photocatalyst was excellent. Almost all *E. coli, S. aureus and P. aeruginosa* could be eliminated, and the antimicrobial performance reached 99.999%. The experiment of active free radical trapping showed that ∙O_2_^−^ and h^+^ were the main active substances in the AgBr/Ag_2_MoO_4_@InVO_4_ photocatalyst. InVO_4_ was compounded with Ag_2_MoO_4_ and AgBr to construct a composite photocatalyst with different heterojunction structures, which facilitated the separation of photogenerated holes and electrons, enhanced the light capture capability, prolonged the light absorption region, restrained recombination of holes and electrons and further improved the photocatalytic performance. Due to its outstanding photocatalytic performance, AgBr/Ag_2_MoO_4_@InVO_4_ shows good prospect in photocatalytic sterilization and environmental pollution control areas.

## Figures and Tables

**Figure 1 nanomaterials-10-01541-f001:**
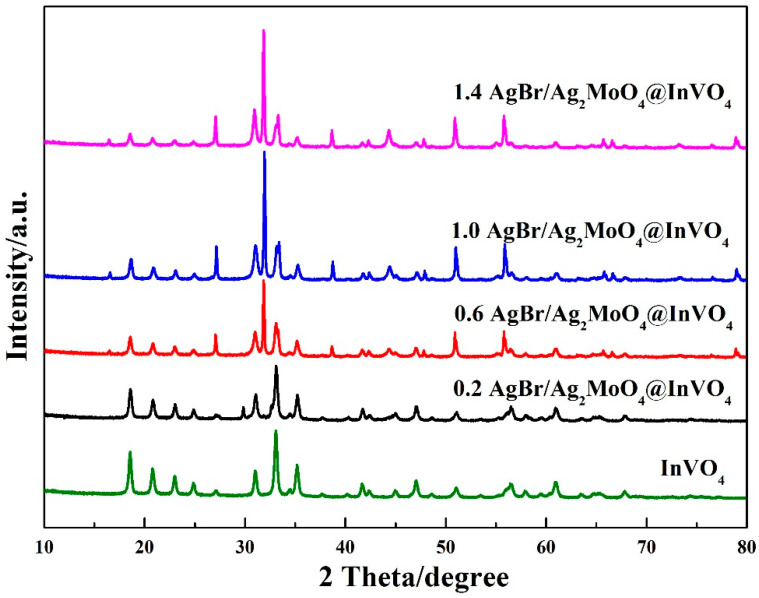
XRD patterns of different molar ratios of AgBr/Ag_2_MoO_4_@InVO_4_ photocatalyst.

**Figure 2 nanomaterials-10-01541-f002:**
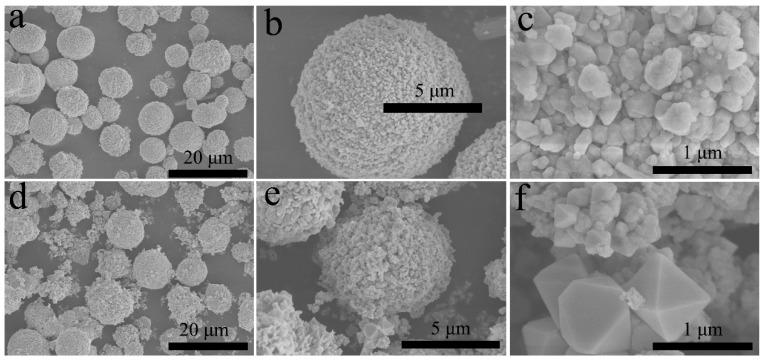
SEM pictures of as-synthesized photocatalysts: (**a**–**c**) 1.0 Ag_2_MoO_4_@InVO_4_, (**d**–**f**) 1.0 AgBr/Ag_2_MoO_4_@InVO_4_.

**Figure 3 nanomaterials-10-01541-f003:**
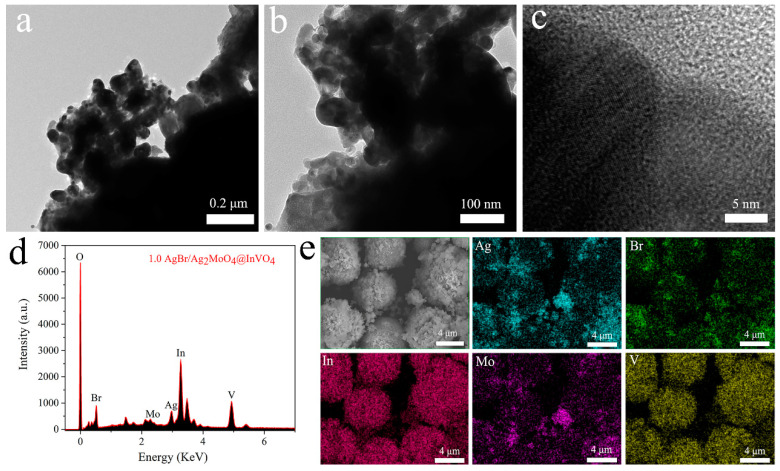
(**a**–**c**) TEM and HRTEM pictures of 1.0AgBr/Ag2MoO4@InVO4 of the photocatalysts. (**d**) EDS pattern and (**e**) EDX elemental pictures of as-prepared 1.0AgBr/Ag_2_MoO_4_@InVO_4_ photocatalysts.

**Figure 4 nanomaterials-10-01541-f004:**
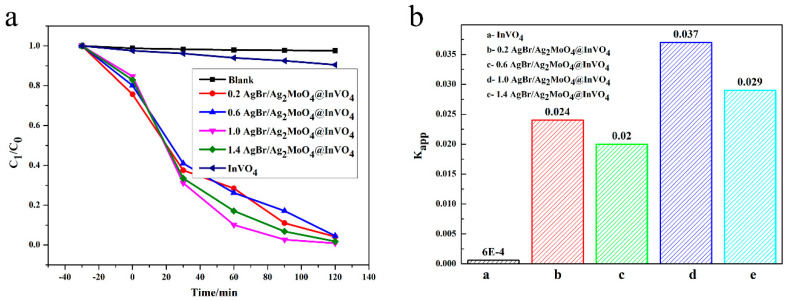
(**a**) Degradation performance of ciprofloxacin (CIP) solution in the presence of different photocatalysts, (**b**) the first-order kinetic constants of CIP degradation for different photocatalysts.

**Figure 5 nanomaterials-10-01541-f005:**
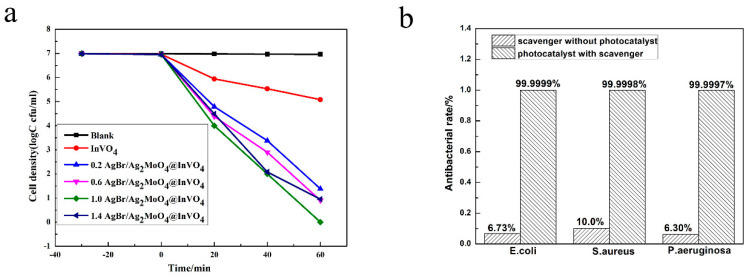
(**a**) Survival curve of *Pseudomonas aeruginosa* in antibacterial test. (**b**) Inhibition rate of 1.0 AgBr/Ag_2_MoO_4_@InVO_4_ photocatalysis to *Escherichia coli*, *Staphylococcus aureus* and *P. aeruginosa* for 60 min under the condition of visible light.

**Figure 6 nanomaterials-10-01541-f006:**
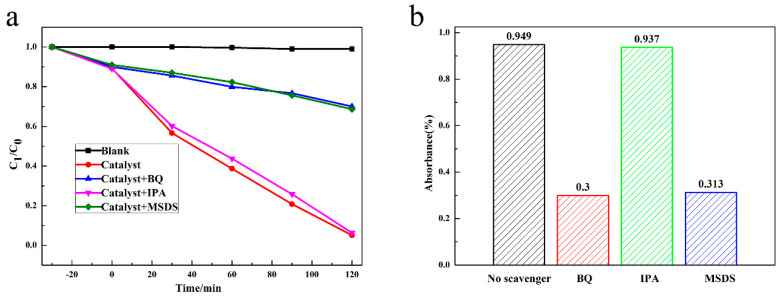
The active matter catching tests for degradation of CIP with 1.0AgBr/Ag_2_MoO_4_@InVO_4_ composite photocatalyst in visible light. (**a**) Degradation curve and (**b**) degradation rate of CIP with different capture agent.

**Figure 7 nanomaterials-10-01541-f007:**
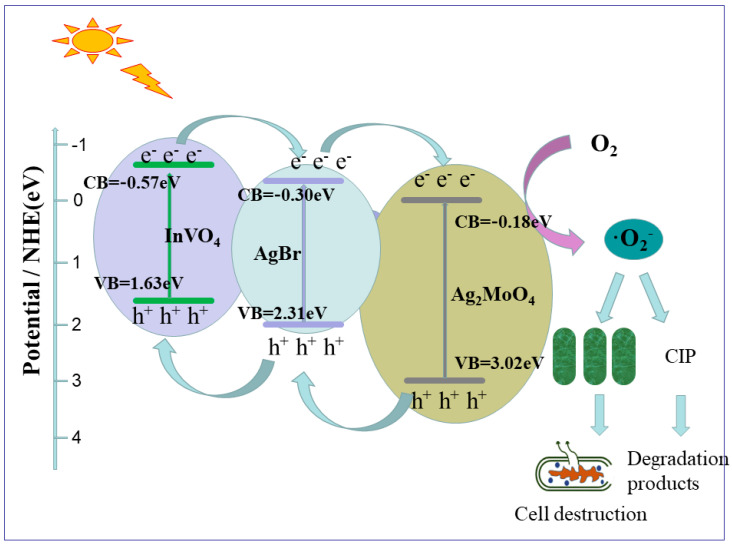
Schematic diagram of AgBr/Ag_2_MoO_4_@InVO_4_ photocatalytic mechanism.

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
