# Peer review of "Fabrication of a Novel AgBr/Ag2MoO4@InVO4 Composite with Excellent Visible Light Photocatalytic Property for Antibacterial Use"

_nanomaterials, 2020, doi:10.3390/nano10081541_

Round 1

Reviewer 1 Report

The paper is interesting and can be published in the journal. However, some points should be addressed before publication. This system is a multi-component and complex system in which the role of each component, impact on the activity, should be clarified as a comparison plot. Furthermore, some related recent references should be discussed in the introduction.

Reviewer 2 Report

The manuscript has potential for publications since a novel material was prepared and applyes on the photocatalytic degradation as well as antibacterial agent, however, there is a lot of work on that, many other nanomaterials have been proposed, especially for CIP removal, so, the novelty should be referred. Also it misses the comparison with other nanomaterials. Is this better as catalysts or as antimicrobial agent than others? Why proposing it? And how about the costs? And toxicity?

The mechanism of CIP degradation was also not explored. In the many other publications on this subject, authors have made an attemp to identify products. 

The proposed mechanism for bacteria? Is dead is also not properly fundamented. Authors only guess.

Round 2

Reviewer 1 Report

The authors have carefully addressed the comments accordingly. It is publishable in the current format.

Reviewer 2 Report

Authors have improved the manuscrip taking the suggestions of reviwers in consideration.